# Novel transdisciplinary methodology for cross-sectional analysis of snakebite epidemiology at national scale

**Gabriel Alcoba**[1,2,3☯]*, **Carlos Ochoa**[2,4☯], **Sara Babo Martins**[2], **Rafael Ruiz de Castañeda**[2], **Isabelle Bolon**[2], **Franck Wanda**[5], **Eric Comte**[5], **Manish Subedi**[6], **Bhupendra Shah**[6], **Anup Ghimire**[6], **Etienne Gignoux**[7], **Francisco Luquero**[7], **Armand Seraphin Nkwescheu**[8], **Sanjib Kumar Sharma**[6], **François Chappuis**[1,9], **Nicolas Ray**[2,4]

1 Division of Tropical and Humanitarian Medicine, Geneva University Hospitals (HUG), Geneva, Switzerland, 2 Institute of Global Health (IGH), Department of Community Health and Medicine, Faculty of Medicine, University of Geneva, Geneva, Switzerland, 3 Médecins Sans Frontières (MSF), Geneva, Switzerland, 4 Institute for Environmental Sciences (ISE), University of Geneva, Geneva, Switzerland, 5 Centre International de Recherche, d'Enseignement et de Soins en Milieu Tropical (CIRES), Akonolinga, Cameroon, 6 B.P. Koirala Institute of Health Sciences (BPKIHS), Dharan, Nepal, 7 Epicentre, Médecins Sans Frontières, Geneva, Switzerland/ Paris, France, 8 Cameroon Society of Epidemiology (CaSE), and Faculty of Medicine and Biomedical Science, University of Yaoundé 1, Yaoundé, Cameroon, 9 Department of Community Health and Medicine, Faculty of Medicine, University of Geneva, Geneva, Switzerland

☯ These authors contributed equally to this work.
* gabriel.alcoba@hcuge.ch

**Data Availability Statement:** Data are available from Ethics Committee (contact via BPKIHS, Nepal, CNERSH, Cameroon, and CCER, Geneva) for

## Abstract

### Background

Worldwide, it is estimated that snakes bite 4.5–5.4 million people annually, 2.7 million of which are envenomed, and 81,000–138,000 die. The World Health Organization reported these estimates and recognized the scarcity of large-scale, community-based, epidemiological data. In this context, we developed the "Snake-Byte" project that aims at (i) quantifying and mapping the impact of snakebite on human and animal health, and on livelihoods, (ii) developing predictive models for medical, ecological and economic indicators, and (iii) analyzing geographic accessibility to healthcare. This paper exclusively describes the methodology we developed to collect large-scale primary data on snakebite in humans and animals in two hyper-endemic countries, Cameroon and Nepal.

### Methodology/Principal findings

We compared available methods on snakebite epidemiology and on multi-cluster survey development. Then, in line with those findings, we developed an original study methodology based on a multi-cluster random survey, enhanced by geospatial, One Health, and health economics components. Using a minimum hypothesized snakebite national incidence of 100/100,000/year and optimizing design effect, confidence level, and non-response margin, we calculated a sample of 61,000 people per country. This represented 11,700 households in Cameroon and 13,800 in Nepal. The random selection with *probability proportional to size* generated 250 clusters from all Cameroonian regions and all Nepalese Terai districts. Our

researchers who meet the criteria for access to confidential data, after approval by these ethical boards and authors' institutions.

**Funding:** This study was funded by the Swiss National Science Foundation (SNSF) (project number 315130_176271, website: http://p3.snf.ch/project-176271) awarded to Nicolas Ray and Francois Chappuis. Gabriel Alcoba was partly funded by MSF (Médecins sans Frontières/Doctors without Borders) (https://www.msf.org/). Rafael Ruiz de Castañeda was partly funded by Fondation Louis Jeantet (https://www.jeantet.ch/). The funders played no role in the study design, data collection and analysis, decision to publish, or preparation of the manuscript.

**Competing interests:** The authors have declared that no competing interests exist.

household selection methodology combined spatial randomization and selection via high-resolution satellite images. After ethical approval in Switerland (CCER), Nepal (BPKIHS), and Cameroon (CNERSH), and informed written consent, our e-questionnaires included geolocated baseline demographic and socio-economic characteristics, snakebite clinical features and outcomes, healthcare expenditure, animal ownership, animal outcomes, snake identification, and service accessibility.

## Conclusions/Significance

This novel transdisciplinary survey methodology was subsequently used to collect country-wide snakebite envenoming data in Nepal and Cameroon. District-level incidence data should help health authorities to channel antivenom and healthcare allocation. This methodology, or parts thereof, could be easily adapted to other countries and to other Neglected Tropical Diseases.

## Author summary

Snakebite envenoming was recently classified as a priority neglected tropical disease by the World Health Organization. Up to five million people are bitten, more than a million envenomed, and around 100,000 victims die, mainly in rural and remote areas of low- and middle-income countries. Snakebite envenoming not only affects victims acutely, but it can also cause long-term disability, disfiguring scars, and heavy economic burden due to treatment costs and inability to work. Previous studies have analyzed snakebite clinical, epidemiological, or socio-economic impacts independently, and little has been done to assess the impact of snakebite in animals and on the livelihoods of the communities that depend upon them.

We present an innovative, holistic, national-scale methodology that includes epidemiology, One Health, economic, and geographic information science approaches into one multi-cluster household survey. We randomly selected 250 sub-district areas from all Cameroonian regions and all Nepali Terai districts, which represented more than 61,000 participants in each country. This methodology could be adapted and implemented in other countries affected by snakebite.

## Introduction

Snakebite envenoming is the second deadliest neglected tropical disease (NTD) [1]. It was added in 2017 to the list of priority NTDs by the World Health Organization (WHO) [2]. Snakebite envenoming has been greatly underestimated as a cause of death and disability in tropical countries. WHO estimated that 4.5–5.4 million people are bitten by snakes globally every year, 1.8–2.7 million are envenomed, and 81,000–138,000 die, with a case-fatality ratio of 4–7% [2]. Likewise, about 400,000 victims remain affected by disability, amputations, skin or muscle scars, brain, eye, or kidney damage, social stigma, and post-traumatic stress disorders [3,4]. As with other medical emergencies, snakebite mortality and morbidity are highly dependent on timely access to appropriate healthcare. Therefore, understanding the geographic accessibility of victims to suitable treatment is crucial to eventually reduce the burden of this disease.

The lack of snakebite epidemiological data was highlighted by WHO in its new Snakebite Envenoming strategy and road map [4,5]. Most of the limited available epidemiological data is mainly hospital-based [6–8], and very few population-based studies on snakebite incidence or mortality have been published. Two meta-analyses attempted to estimate the global incidence of snakebite [6,7], but they were also mostly based on hospital records and small, district level, surveys rather than national studies. Country-wide snakebite incidence studies using spatial epidemiology, as done in Bangladesh [9] and more recently in Sri Lanka [10], seem to be the best approach to understand snakebite impact at national and sub-national scale. Such large scale studies are still rare or are limited to only one aspect of the disease, namely mortality, as seen in the snakebite component of the Indian million-death study [11]. Other studies of snakebite epidemiology at national scale have been done retrospectively using hospital records or long-term governmental surveys [8,12–14], but their scope was still mostly limited to incidence or mortality, and did not include accessibility, health economics, nor One Health components.

Recent studies have allowed to estimate human vulnerability to snakebite envenoming globally, by using proxy data on risk factors such as healthcare gaps, unavailability of antivenoms, and presence of medically important snake species [15], or through mathematical models [16], but without a direct measure of snakebite incidence and mortality. Additionally, new digital health applications have shown that it is partially possible to estimate areas of high snakebite incidence and identify populations at risk through citizen science [15,17,18] in countries where social media, citizen forums, and scientific online platforms are used to report snake-human encounters and map their geographic distributions [17,19].

Snakebite envenoming can also affect domestic and livestock animals, having a direct impact on their productivity and mortality, which can significantly affect the livelihood of rural communities. This is a highly neglected issue, and little is known about snakebite impact in animal health and production [20]. A comprehensive assessment of the societal impact of snakebite envenoming in affected communities needs a systemic, One Health approach taking into account its double socio-economic impact linked to human health and livelihood losses [21].

Our research focused on Nepal and Cameroon, two countries with reported snakebite hyper-endemism that makes them credible representatives of the snakebite crisis in sub-Saharan Africa and South Asia. With our Nepalese, Cameroonian, and Swiss partners, we designed and structured the "Snake-Byte" project to integrate three aims: (i) to quantify and map the impact of snakebite on human and animal health, and on livelihoods, (ii) to develop predictive models for medical, ecological and economic indicators, and (iii) to analyze geographic accessibility to healthcare. The achievement of these interconnected objectives required the development of a novel transdisciplinary methodology for snakebite epidemiology and primary data collection.

Surveying whole populations at national scale is practically impossible, except for expensive population demographic censuses. However, the recent availability of satellite and aerial imagery has facilitated geographically assisted population sampling, proving to be a valuable tool for epidemiology and global health [22,23]. Consequently, we assessed the range of methodologies that would insure optimal efficiency, validity, and representativeness of data collections in large populations [24].

This paper describes the methodology we developed after the evaluation of multiple epidemiological approaches and practical methods, in order to study several aspects of snakebite (and snakebite envenoming) at the national level and to estimate snakebite's societal impact from a transdisciplinary perspective. We adapted these methods to our research objectives in Nepal and Cameroon, and hereafter provide the necessary elements for their application in

similar epidemiological studies on snakebite in other countries. In the Methods section, we present how the choice of the research methodology was done, and in the Results section, we describe how the selected research methodology was developed.

## Methods

### Ethics statement

Ethical approvals were obtained in the three countries, in Cameroon from the *Comité National d'Ethique de la Recherche pour la Santé Humaine* (CNERSH, No. 2018/09/1208) on 14 September 2018, in Nepal from the National Health Research Council (NHRC Reg.no. 585/2018) on 10 October 2018, and in Switzerland from *Commission Cantonale d'Ethique de la Recherche scientifique* in Geneva (CCER and Swiss Ethics Registry No. 2018–01331) on 23 October 2018. Written informed consent were obtained from all participants, or their legal representative, depending on legal age in each country.

### Evaluation of epidemiological methods

Previous comparative analyses of epidemiological methods in household surveys for NTDs and epidemics have shown great convergence towards a trade-off between simplicity and accuracy [25–27]. The cross-sectional multi-cluster survey, has been a frequently used survey methodology in humanitarian and resource-limited contexts, especially for crude mortality rates, malnutrition rates, and vaccination-coverage. However, this is not the sole survey design and methodologies have evolved considerably in the past 40 years in order to assess a range of topics, from shelters or access to safe water to the prevalence of hookworms in malnourished children [28,29].

Similarly, the recommendations concerning the number of sampling units (individuals or household) per cluster (village, district) have greatly changed over recent decades. The sample size of field surveys has shifted from the classic 30 clusters × 30 households (i.e. 900 households) to the smaller 30 × 15 and then 30 × 7. These three sample sizes showed very similar accuracies, precisions and inter/intra-cluster correlations, and reliabilities regarding surveys with frequent outcomes or high prevalence (above 5–10%), such as vaccination coverage or child malnutrition surveys [28–30].

Obtaining a representative national prevalence or an annual incidence of rarer events like snakebite envenoming, with an expected incidence <1%, required using a substantially larger sample size that allowed to acquire reliable incidence data, including an additional margin for sub-analyses (stratification). At the same time, the sample size should be commensurate with the time frame and resources available for the intended project. To determine our sample size, we analyzed methodologies previously used on snakebite epidemiology studies [9,10] and consulted expert survey epidemiologists at Epicentre-MSF (*Médecins Sans Frontières*, Doctors Without Borders). We also assessed methodologies employed to study other NTDs and health indicators in tropical, low- and middle-income countries, as well as in rural and remote populations [31–33]. Additionally, we compared standardized methods for multi-cluster surveys: the national level Multiple Indicator Cluster Survey (MICS) by the United Nations Children's Fund (UNICEF) [34], the Demographic and Health Surveys (DHS) programme by the United States Agency for International Development [35,36], as well as MSF-Epicentre's survey method for large and remote populations, which also included a review of multi-cluster surveys [25].

To conduct the joint human-animal survey, we have followed the approaches proposed to investigate simultaneously humans and animal populations in a One Health perspective in previous studies for zoonoses and other associated topics. Sample size calculation in One Health studies is often based on one population [37–39]. In this study the sample size was powered to estimate the incidence in humans, with the domestic animal sample incidental to that

for humans. Although differences in the number and type of domestic and livestock animals per household across regions in each of our study countries are expected, animals are a fundamental source of income in rural communities, and it could be anticipated that most sampled household in the sampling frame will own domestic animals [40].

## Cluster selection

Cluster selection methods strongly depend on the aim of the survey. The selection can be done randomly or deterministically [41,42] according to the problem to be confronted and the type of population affected, and in some cases clusters might need to be created instead of selected [43]. Regardless of the method chosen, in a multi-cluster survey, the selection has to provide the appropriate basis for the secondary units' selection, usually household.

We analyzed several studies surveying populations at different geographical scales, countries and logistic conditions, trying to extract methods that could be applicable to our specific settings in Cameroon and Nepal at both national and sub-national scale. Some of those studies in Lebanon, the Democratic Republic of the Congo, Rwanda, and Iraq used or suggested the use of weighed selection, mainly via *probability proportional to size* [44–47]. The supporting software for the selection process varied from freeware such as *Google Earth* [41,46,48], *R* [47,49,50] and *QGIS* [51,52] to commercially licensed such as *ArcGIS* [43]. Most of these studies aimed at proposing survey methods optimized for their specific conditions, and therefore less replicable in other settings.

## Household selection

We evaluated the main household sampling methods: systematic random sampling vs. cluster random sampling, the latter using either *random walk*, *compact element* or *spatial sampling*. First, systematic random sampling, using fixed intervals from the entire population, is statistically precise but extremely costly, time consuming, and unrealistic for nationwide field surveys. This method has been used in settings like refugee camps or small populations, but gradually abandoned in favor of the more efficient cluster sampling [25].

Second, cluster random sampling (CRS) uses clusters such as villages or towns of comparable size, selected from a census list based on *probability proportional to size*. Then within each cluster, household-level selection could be performed using various methods. One of those methods is *random walks*, which has been used in earlier studies to ensure that surveyors advance in a random direction and select random houses along the way. Although practical, this method has showed substantial spatial selection bias [25], often due logistical oversimplification or the surveyor's personal preferences. Third, CRS with *compact element* method divides population units into grids of squares, allowing to select them randomly on paper or digital maps [53]. This *compact element* method ensures objectivity by reducing the risk of spatial bias through random selection of the movement direction [27]. Finally, the more recent CRS with *spatial sampling* methods employ geographic information systems (GIS) for randomly selecting households with the help of satellite imagery, which has proved to be as effective and more objective than previous methods while further reducing selection bias [23,26,43,54] (see Table 1).

## Spatial sampling methods for random household selection

Previous studies have shown that GIS random selection methods reduce selection bias with respect to traditional cluster methods [43], highlighting the importance of free satellite imagery. Even if many online map services offer free and recent satellite/aerial images, the imagery for some regions may not be up-to-date and consequently some populations might be neglected. An alternative is to use more recent commercial satellite imagery [55–57]. Even

**Table 1. Comparison between common sampling methods.**

| Method | Advantages | Disadvantages | Ref. |
|---|---|---|---|
| Systematic Random Sampling | • High precision<br>• No selection bias | • Expensive<br>• Time-consuming | [25] |
| Cluster Random Sampling (CRS) with random walk | • Easy to perform by shortly trained surveyors<br>• Fast | • Severe risk of selection bias | [25] |
| CRS with compact element | • Reduced risk of selection bias vs. random walk<br>• Objective cluster selection | • Moderate risk of selection bias<br>• Risk of clustering bias | [27,53] |
| CRS with spatial sampling | • Better random point selection | • Low risk of selection bias<br>• Low risk of bias excluding sparsely populated areas | [23,26,43,54] |

though that option is usually too expensive for large-scale surveys, it might be the only alternative in some situations, such as post-disaster settings [43]. Other studies have used different approaches, such as crowd-sourcing, to collect household geographic locations [54], but then only covering relatively small local areas (1,600 km$^2$). *Google Earth* satellite imagery was also used, in combination with specialized software, to geolocalize every single household in Lilongwe, Malawi [23], which generated about 18,000 points in a relatively small area during a period of three months. Such an exhaustive method would be difficult to extrapolate at national scale in larger countries.

Previous studies have also used multiple combinations of methodologies and software to generate random points (e.g. *R*, *Stata*, *or GPS Sample*), and to map and select households either digitally (e.g. *Google Earth*) or physically with GPS units, in order to randomly select the allocated points for each cluster [23,43,49,54,56,58]. Several additional methods have been used to complement random samplings, such as generating more random points than necessary as a precaution in case of no answer or the absence of an eligible household in the location selected [43]. In previous studies, the surveyors were required to select or replace households by heading north of a GPS point [25], find all the households closest to a point and select one randomly [59], or find a suitable replacement within a determined radius area [43]. Our methodological approach aimed at a simplified household selection process that is adaptable, efficient, easy-to-replicate at different geographical scales, and can be run completely within a free and open-source software such as *QGIS* [52].

## Geographic guidance of field teams

Following the household selection, two main methods have been used to physically locate the selected households [53,60,61]. Some studies have employed dedicated GPS devices, while others have opted for the use of electronic tablets or personal digital assistants with geolocation capabilities, to combine that functionality with the ones provided by other applications to collect survey data and images, for instance. For that purpose, we tested several Android applications (*OruxMaps*, *Map.me offline*, *GPX Viewer*, and *GuruMaps*) trying to find a balance between usability (for surveyors in the field) and functionality (offline maps, data uploading, downloading and sharing, live guidance, etc.).

## Results

### Sample size calculation

After exploring the alternative methods described in the previous section and Table 1, and their advantages, we opted for a multi-cluster sampling method, CRS with spacial sampling,

guaranteeing a representative sample of the target population and minimizing sampling error. We followed the most comprehensive methodologies for large populations sampling, such as MICS, DHS, and increased it for sub-analyses (clustering). After testing three similar calculation formulae (MICS, DHS, and OpenEpi), we selected Open Epi (http://www.openepi.com/SampleSize/SSPropor.htm) because it is easily accessible, open-source [62,63], and provides a clear, online sample size calculator, applying the following formula:

$$n = \frac{[DEFF * N * p(1-p)]}{\left[\left(\frac{d^2}{Z^2_{1-\frac{\alpha}{2}}}\right)(N-1) + p(1-p)\right]} \tag{1}$$

Where, $n$ is the sample size, $N$ is the total population size, $p$ is the hypothesized percentage of occurrence, $d$ is the confidence limit value for $p$, $DEFF$ is the design effect (degree of variance due to cluster-sampling), and $Z$ is the number of standard deviations any observation is away from the mean in the Standard Normal distribution for a given $\alpha$ value.

The hypothesized cross-sectional prevalence of snakebite in the past 12 months (i.e. annual incidence rate of snakebite) or $p$ "percentage of occurrence" in the formula above was based on previous surveys and hospital records [6,7] coming from the same countries or regions and reporting annual incidence rates per 100,000 people: in Benin (216–653), Congo (125–430), Kenya (150) and Nigeria (48–603), and hospital record estimates of 75–200 in Cameroon, and 100–120 in Nigeria, with case-fatality rates ranging from 1% to 11%. Similarly, in the Asian-Pacific region, annual incidence rates per 100,000 people from hospital-records ranged from 3–18 in Australia to 66–163 in India, 400–450 in Malaysia, and 215–526 in Papua New Guinea. For Nepal, a regional random survey in 2001 found an incidence of 1,162/100,000, but the study purposefully targeted villages of presumed high incidence [64]. Two Nepali studies reported very high hospital snakebite admission rates [65,66], but no community-based district or provincial incidence.

## Incidence estimates for sample size calculation

Based on the previously mentioned values, we estimated a minimal "conservative" incidence of 100 snakebite events per 100,000 people per year (0.1% prevalence, plausible confidence limits ± 0.05%) for both countries, allowing possible over-powering, rather than the contrary. The confidence level was set at 99% to maximize the precision of the estimates. The $DEFF$ (design effect) was increased to 2.0 (usual MICS and DHS surveys: 1.5–1.7) to compensate any potential inter-cluster heterogeneity, as recommended by MICS/DHS. This calculation resulted in a sample of 52,885 individuals. Likewise, we added a *margin for non-response* of 15% that totaled a sample of 61,000 participants per country.

## Household sample size

Given the average number of people per household in our study countries (5.2 in Cameroon and 4.4 in Nepal), and the total sample size of 61,000 persons, we obtained a household sample size of 11,700 for Cameroon and 13,800 for Nepal. These numbers were 2–4 times higher than those required for a simple national incidence figure with a confidence level of 95%. Additionally, setting a high $DEFF$ value and geographically covering most of the available population reinforces the reduction of selection bias. For a given number of participants, increasing the number of clusters (thus reducing number of participants per sample) usually increases the level of precision. However, increasing the number of clusters can have a severe financial impact on the study in large countries (e.g. travel time and fuel costs). To optimize the geographical representative coverage, we targeted a 5:1 ratio, i.e. 250 clusters with a sampling of

approximately 50 households per cluster (11,700/250 = 47 in Cameroon, and 13,800/250 = 55 in Nepal).

## Cluster selection

Representativeness of all large administrative areas (regions) was assured through the selection of 250 clusters in each of the study countries, with probability proportional to their population size. Respecting the recommendations by UNICEF and the national Ministries of Health, we used an equitable representation of all regions in the two countries, thus insuring an appropriate balance of political and cultural areas [34]. Our aim was to include all 10 regions of Cameroon and all 7 provinces of the Terai lowlands in Nepal. We expected to maximize geographic coverage by including also areas at risk of exclusion if simple random surveys were used, namely remote and rural areas, while keeping costs and logistic constraints to a reasonable level, depending on salaries, number of clusters, distance, vehicles rented, fuel, and materials. Considering that snakebite is primarily a rural health concern and based on previous research [15,67], we opted for filtering out the clusters in urban or highly populated areas. This allowed us to select our 250 clusters out of more than 1200 available in each country (a 1:5 ratio), which was deemed sufficient for equitable geographic representativeness.

The selection of clusters was based on the sub-district administrative divisions in each country: the *Gaunpalikas* or rural municipalities (most commonly known by their former designation Village Development Committees, VDC) in Nepal and the Health Areas (HA, in French *Aire de Santé*) in Cameroon. These divisions, although very different in size within and between countries, were the smallest defined administrative division that met our requirements for population size. In agreement with the goal of nationwide representativeness for cluster selection, we opted for a random selection of clusters with PPS weights and the use of the software *R* for replicability.

## Geographic representativeness

Based on the methods described above, our survey included: 1) all administrative regions in Cameroon and all the Terai region of Nepal; 2) randomly selected clusters, HA in Cameroon or VDC in Nepal; and 3) random geospatial sampling and individual selection of household. We based our selection process on two tables containing all levels of administrative divisions of the country down to cluster level. The first source for the tables was the attribute tables of the polygon shape files used to plot the clusters' maps, which are available from international humanitarian organizations such as the United Nations Office for the Coordination of Humanitarian Affairs (*UNOCHA)* [68]. The second source was national institutions, such as geographic or health institutes with access to population data. These tables included, in addition to all administrative divisions, values on the area of the clusters and the population within.

For the exclusion of urban and highly populated areas from our survey sample, we first filtered out all the clusters clearly designated as 'urban' or 'city'. We then kept clusters with populations between 2,000 and 20,000 people. The upper population limit helped to avoid large conglomerations where snakes might be rare or absent. The lower limit was set for logistic and practical purposes to avoid clusters where the population might be too scattered or where the number of households were too low to make a meaningful random selection. In our case, and depending on the country's average household size, a 2,000-people cluster included 385 households in Cameroon and 454 households in Nepal. This meant that to fulfill our sample sizes, one out of each 8.2 households could be selected (12.2% chance).

Additionally, in Nepal, we specifically targeted the Terai region (southern plains) where the vast majority of the rural Nepali population lives. This decision was based on expert advice

and Ministry of Health data showing that snakebite incidence in the other two northern eco-zones (high hills and Himalayas) is expected to be very low [65,66], due mostly to the high altitudes and the much lower human population density. In Cameroon, we agreed with local experts to exclude a buffer zone of 30 km all along the border with Nigeria for security reasons.

Next, we selected the clusters in each country in a random reproducible manner, using the statistical software *R* [50] (see S1 Script). For this selection, the clusters were weighed by PPS, in relation to the total available population (after filtering). We also assured the uniqueness of the selected clusters by combining district and cluster names as single units. This prevented confusing clusters with the same name in different districts.

### Random household selection

After the clusters were selected and mapped in *QGIS*, a populated location inside each cluster was chosen by visually inspecting the terrain with the help of any of four satellite imagery backgrounds (*Google Hybrid*, *Bing Satellite*, *ESRI Satellite* or *Yandex Satellite*) provided by the free plug-in *QuickMapServices* (version 0.19.10.1) in *QGIS*. Switching between satellite image backgrounds helped to compare and select the most recent or sharper image, free of cloud cover. At this stage, it was important to consider the potential logistic difficulties for the field surveyors, including walking distances and vehicle accessibility.

The individual selection of households in the chosen locations entailed both a random component and a user decision process. Depending on the shape of the selected population and the spatial distribution of the households, we created one large or several small areas covering all the households (using *QGIS* '*Random Points in Extent*' function). These areas were filled with randomly distributed points at an approximate density of 500/km². In average, that allowed for a random spread of the points and a good coverage in case the households were sparsely distributed. The same *QGIS* function allowed setting the minimum distance between points. We set that value at 20m, partly to avoid overlapping, and partly to consider the empirical precision of the GPS units in the tablets we used during the actual survey (ranging between 5 and 20m). This distance facilitated pinpointing nearby buildings with the GPS application and also allowed us to create a non-intersecting 10m buffer around each point (Fig 1). This

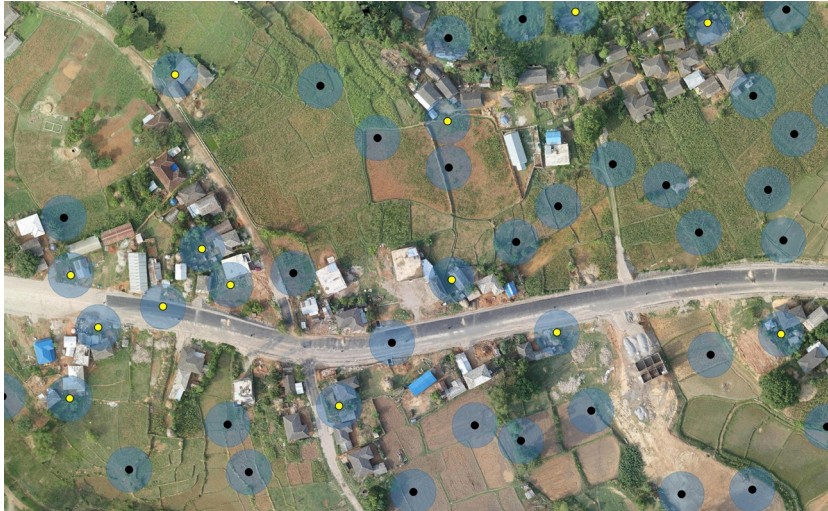

**Fig 1. Example of household selection using random points and 10 m-radius buffer area.** The yellow dots exemplify the selected households. The size of the dots has been increased for clarity. Sample image from http://openaerialmap.org with similar viewpoint to the ones used in the methodology (licenced CC BY 4.0).

buffer served as a range inside which a household could be assigned to a point (if any part of the building was inside the buffer, see Fig 1). The buffer settings (20% transparency) were saved as a style to be re-applied at each repetition. Next, we proceed to the visual selection of household using the '*selection by single click*' tool in *QGIS*. As part of the user decision process, we avoided selecting buildings that were evidently not households. This process could be additionally supported by using satellite imagery backgrounds with commercial labels, or local expertise. In contrast to other approaches previously discussed (see Methods), our methodology allowed to select (and made ready for field work) ca. 25,500 in less than two months of work (about 37 workdays). However, the actual selection of households (considering an already streamlined methodology) took in average 15 minutes for each of the 500 clusters. This should allow scaling up for larger areas or quicker data availability.

In addition to the selected households and based on pilot field trials, we also developed a standard operating procedure (SOP, available upon request) for the replacement of locations in case the chosen ones were not actual households or were empty. This was later justified by our survey results from Cameroon, where in average 75.9% (95%CI: 73.7–78.4) of the locations reported to be correctly identified households and 24.1% (95%CI: 21.6–26.6) were buildings described as having other purposes, being non-existent or abandoned (Fig 2). This data was only available for Cameroon, following a late improvement of the questionnaire.

As a rule, we selected about 10% more buildings than the calculated quota for each cluster, i.e. about 52 (47 + 5) for Cameroon and 60 (55 + 5) for Nepal. This excess can be removed (and in our case, it was) at a later stage of the data preparation or be appropriately saved for later use as a backup. Regarding the efficiency of the randomly generated points to hit actual households, this method was based on creating an excess of random points (between 1,000 and 10,000) covering completely the inhabited area (population), which made it always possible to find the 50–55 households needed.

After the household selection, the information was saved and exported as *.shp* and *.kml* files together with geographic coordinates and unique identifiers, which were later used for the

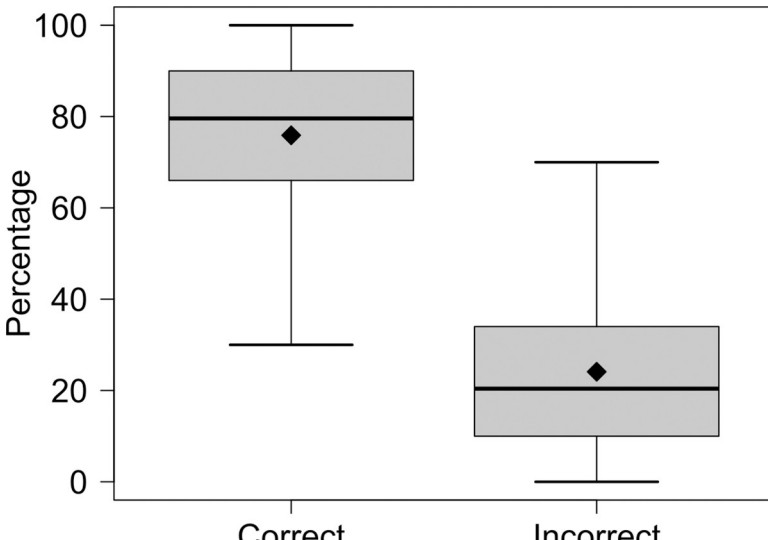

**Fig 2. Percentage of household selection success in clusters, via satellite imagery in Cameroon.** Results of correctly and incorrectly selected household using high-resolution satellite background images in *QGIS*. The thick lines indicate the median, the diamonds display the mean, the box delimits the interquartile range (IQR) and the whiskers give the minimum and maximum values inside the 1.5 IQR.

surveyors' guidance on the field. For that purpose, we adopted the application *GuruMaps Pro* V 2.1.9 [69] (formerly known as *Galileo offline maps* at time of deployment), together with Samsung Galaxy Tab A6 tablets (SM-T585) with Android mobile operative system (version 6.0 Marshmallow). This application offered easy importing, exporting, storage and manipulation of geolocations in GPX and KML formats. It included online/offline maps, voice-assisted guidance, and track recording possibilities. Tablets were enabled with mobile data, which allowed online data synchronization and remote assistance when needed.

## Data collection and e-questionnaires

The household survey was conducted by mobile teams of one to three enumerators and one coordinator, guided by a local community health worker with previous experience in community-based health campaigns (e.g. vaccination campaigns, or mosquito net distribution). Each team was in a car or on motorcycles, depending on road and logistic conditions.

In order to provide an annual incidence and to limit recall (memory) bias, human and animal victims were included only if bitten during the past twelve months (using official holidays and festivities as key time limits). Informative material about the project was distributed and written consent was obtained from every participant for the general demographic questionnaire (No.1), as well as consent for more detailed questionnaires on the victims' medical information (No.2), for photographing snakebite wounds (No.3) or the environment where the bite occurred (No.4).

Concerning the length of the survey, it was necessary to find a balance between the number of multidisciplinary questions and the practicality of it during the fieldwork. A long and complex questionnaire could reduce the interest and cooperation of survey participants and would make the data cleaning and analysis more difficult. Therefore, during the development phase, we extensively tested multiple versions of the questionnaire to achieve a final result that could be recorded in less than 10 minutes (household with no victims) and up to 30–45 minutes (household with human and/or animal victims).

Regarding the survey development, we used the *KoBoToolbox* [70], a suite of tools to create and deploy surveys, collect data and do basic analyses and visualization. One of its principal advantages was to allow flexibility in the type of input (numerical, categorical, conditional, and GPS points) through its excel-type XLS/*Enketo/ODK* forms, with a simple coding system (xlsform.org/). This tool was used jointly with the open source mobile application *KoBoCollect*, which is based on the *OpenDataKit* software and is used for primary data collection in humanitarian emergencies. It allows running surveys in the field with help of electronic tablets or smartphones, and securely encoding and storing them both locally and remotely in a server.

## Scope of data collection

The household questionnaire aimed primarily at investigating snakebite incidence in humans and animals, with and without envenoming, during the previous year. The baseline questionnaire covered demographic composition, housing characteristics, and numbers and species of animals kept. In case of snakebite cases (human or animal), the questionnaire requested further information on the victims, such as snakebite context (e.g. location, date, time of the day), snake characteristics (including identification in a photo album), diagnosis and health consequences (e.g. symptoms, clinical outcomes), health-seeking behaviors and treatment (e.g. medical, veterinary or traditional care, antivenom administration and means of transportation), and costs (S1 and S2 Kobotoolboxes). Likewise, questions on health expenditure and productivity losses due to absenteeism were included in the questionnaire, drawing on previous studies on the economic impact of snakebite in other countries [71]. Additionally, we aimed to

collect data following the Poverty Probability Index (PPI) scorecard on PPI indicators for both countries.

Regarding the cases of snakebite on animals, we developed a specific SOP to clarify the animal questionnaire completion process in cases where, in a selected household, more than one animal was bitten in the same episode (i.e. grouped cases, available upon request). The questionnaire also integrated questions on snakebite morbidity and mortality of cattle, poultry, other production species and pets, associated production losses and any treatment costs, as well as some ecological dimensions, such as snake species and habitat, and contact/accident environment. This was considered relevant for assessing risks and developing preventive solutions. The questionnaire for animal victims was built in agreement with a recent scoping review on the impact of snakebite on domestic animals [20], which helped identify snakebite envenoming clinical signs and animal health consequences, and define the criteria for the diagnosis of snakebite in animals. Specific questions on the reasons why the interviewee concludes the animal was bitten by a snake and on the clinical signs observed were added in the questionnaire, which helps categorize snakebite in animals as confirmed, probable or suspected during data analysis. By integrating and collecting data on human, animal and environmental domains, this study aligned with the COHERE standards for One Health study design [72]. This allowed to address specific research questions in the project, namely the impact of snakebite in the livelihood of affected communities, while also providing a first understanding of incidence rate of snakebite cases in animals. The questionnaires were internally and externally reviewed, and tested in the field prior to the study start.

## Discussion

### Originality

This novel methodology for the study of snakebite epidemiology and the associated burden integrates national and sub-national incidence, spatial epidemiology, human and animal health impacts and socio-economic consequences in two hyper-endemic countries, with a One Health perspective. These elements are key in the design and implementation of public health interventions, such as preventive measures and optimized supply of antivenom and acute medical care. The originality of this approach also extends to the possibility of mapping epidemiological risk of snakebite envenoming and developing predictive models of geographic accessibility to healthcare, grounded in a solid foundation of primary data. This methodology is aligned with one of the pillars of WHO's Strategy (Health systems, including surveillance) for Prevention and Control of Snakebite Envenoming [4,5].

### Study feasibility

Other studies have taken different approaches for every step in the process of selecting primary and secondary sampling units. According to our specific set of conditions (type and field of study, scale, duration, etc.), our combination of random selection methods, freely available software and data sources, and data management hardware and software proved to be flexible and robust. Our approach has the advantage of being adaptable to variable conditions (divisible into 2–4 modules, purely human snakebite incidence, animal snakebite, geo-health, or health economics aspects), applicable at national or local scales, and include *probability proportional to size* or any other weighing factor of interest, or none. It also offers the possibility of modifying the general sample size selection factors and the household selection parameters, according to the requirements and context of each study. In addition, our field methods combine hardware and software that allow both high independence for the field teams and the possibility of survey and location changes in real time. Finally, for the purpose of generalizability,

**Table 2. Structural modules for developing snakebite multi-cluster random surveys.**

|  | Ideal scenario | Minimal scenario |
|---|---|---|
| Representativeness | • National (all large administrative areas) | • Regional |
| Sample size | • Large: > 0.3% of the population<br>• (e.g. n = 60,000 people for N = 20,000,000 country population)<br>• Allows district rates and impact analyses | • Minimal: approx. 0.05% of the population<br>• Allows national incidence rate and few sub-analyses |
| Sampling method | *Spatial sampling*:<br>• random GPS points with preselected density of points<br>• up-to-date satellite image using multi-sourcing | *Compact element sampling*:<br>• random selection of grid cells from an updated map (avoid random walk method). Preselect grid boxes |
| Time frame | • Field work: several months<br>• Analysis: Retrospective on annual incidence (past 12 months) with clear cut-offs (calendars dates) | • Field work: 1–2 months<br>• Analysis: retrospective on annual incidence (past 12 or 24 months, to increase episodes) |
| Logistics | • 3–4 mobile teams traveling by car<br>• 3–4 surveyors per car<br>• Tablets with data collection tools.<br>• Drivers familiar with the area, guided by GPS to preselected households | • Mobile or fixed teams, e.g. community health workers (CHW) from each preselected cluster with mobile phones<br>• Tablets/paper data forms<br>• Drivers helped by well-designed map grids<br>• Drivers/CHW familiar with the area |
| Field work costs | • Depending on local logistics and number of clusters | • Depending on distance, local logistics, and number of clusters |

we summarize in Table 2 the required conditions and methods for a wide-ranging or a minimal scenario for a snakebite cluster random survey.

## Study limitations

We encountered very few limitations in the methodology design as well as in its implementation in the field. One was the relatively high costs linked to personnel and transportation during the field data collection (about 4 months), which was expected from a national scale survey. However, this cost was reduced through a balanced cluster selection, and targeted transport with real-time GPS guidance towards the selected locations.

Evidence from advanced stages of the household selection process showed that using population size thresholds during cluster selection, not always led to choosing clusters with the desired properties. If population sizes and cluster areas are known, using the population density or an index thereof might provide a more precise filter to set apart rural and urban areas.

Our approach should strongly reduce selection bias by using randomization at both levels: clusters (PPS) and households (random spatial sampling). Yet, minimal bias due to exclusion of buildings that did not look like households might be possible. Recall bias is a frequent type of bias in this type of cross-sectional study, using participants' retrospective memory on a period of twelve months. However, the snakebite accident is such a traumatic event, that it is much better remembered than most health events, thus reducing the risk of recall bias [73]. Minimal selection bias could result from the exclusion of purely urban areas.

Because the domestic animal sample is incidental to that for humans, animal population characteristics, such as the heterogeneity in the distribution of the animal populations, might not be fully captured. This might lead to some bias and it is a common limitation of surveys conducted with a One Health approach where multiple species, including humans, are sampled.

## Conclusion

This novel methodology for measuring the epidemiological impact of snakebite envenoming at national and sub-national levels in a holistic, transdisciplinary manner proved feasible, flexible, and robust for two very different snakebite hyper-endemic countries (Cameroon and

Nepal). The proposed methodology also allowed us to survey snakebite in domestic animals, an understudied area in terms of snakebite incidence and impact [20]. Outcomes of the survey should therefore provide an important stepping stone for the design of future studies focusing on snakebite in a diversity of domestic and livestock animals at regional or national level.

We hope this methodology will contribute to bring nationwide data on snakebite envenoming into the spotlight, confirming its importance as a major NTD of public health importance, and raising scientific, political, and public awareness, in line with WHO objectives [4]. This methodology addresses WHO's call for large-scale, comprehensive epidemiological research, focusing on collecting human and animal primary data at the community level, and using a recommended One health approach [21,74]. It could also be generalized to assess national or sub-national snakebite burden in other countries, with the necessary community engagement and political support. Moreover, this methodology might be applied to other under-reported Neglected Tropical Diseases that could also benefit from a One Health approach. Likewise, the eco-epidemiological data resulting from applying this methodology could provide comparable national baselines, guiding progress towards WHO's national and global objectives of snakebite mortality and morbidity reduction by 2030.

## Supporting information

**S1 Script. Cluster random selection.**
(R)

**S1 Kobotoolbox. Questionnaire for Nepal in English and Nepali (for calendar).**
(XLS)

**S2 Kobotoolbox. Questionnaire for Cameroon in French and English.**
(XLS)

## Acknowledgments

The authors would like to thank all the members of the Cameroonian Ministry of Public Health and the Nepalese Ministry of Health and Population who allowed this study, and the three Ethical review boards in Cameroon (CNERSH), Nepal (NHRC) and Switzerland (CCER). We thank Dr. M. Herrera (Instituto Clodomiro Picado, San José, Costa Rica) and A. Nickerson (Institute for Global Health Sciences, University of California, San Francisco, USA) for reviewing the e-questionnaire and for their constructive comments. We thank Dr A. M. Durso (Department of Biological Sciences, Florida Gulf Coast University, Ft Myers, USA) for his help developing the snake photo album. We thankfully acknowledge the field teams that have conducted the survey in the two countries.

## Author Contributions

**Conceptualization:** Gabriel Alcoba, Carlos Ochoa, Sara Babo Martins, Rafael Ruiz de Castañeda, Isabelle Bolon, Franck Wanda, Eric Comte, Etienne Gignoux, Armand Seraphin Nkwescheu, Sanjib Kumar Sharma, François Chappuis, Nicolas Ray.

**Data curation:** Gabriel Alcoba, Carlos Ochoa.

**Formal analysis:** Gabriel Alcoba, Carlos Ochoa.

**Funding acquisition:** François Chappuis, Nicolas Ray.

**Investigation:** Gabriel Alcoba, Carlos Ochoa, Sara Babo Martins, Rafael Ruiz de Castañeda, Isabelle Bolon, Franck Wanda, Manish Subedi, Bhupendra Shah, Anup Ghimire, Armand Seraphin Nkwescheu, Sanjib Kumar Sharma, François Chappuis, Nicolas Ray.

**Methodology:** Gabriel Alcoba, Carlos Ochoa, Sara Babo Martins, Rafael Ruiz de Castañeda, Isabelle Bolon, Franck Wanda, Eric Comte, Anup Ghimire, Etienne Gignoux, Francisco Luquero, Armand Seraphin Nkwescheu, Sanjib Kumar Sharma, François Chappuis, Nicolas Ray.

**Project administration:** Sara Babo Martins, Franck Wanda, Eric Comte, Armand Seraphin Nkwescheu, Sanjib Kumar Sharma, François Chappuis, Nicolas Ray.

**Resources:** François Chappuis, Nicolas Ray.

**Software:** Gabriel Alcoba, Carlos Ochoa.

**Supervision:** François Chappuis, Nicolas Ray.

**Validation:** Gabriel Alcoba, Carlos Ochoa, Sara Babo Martins, Rafael Ruiz de Castañeda, Armand Seraphin Nkwescheu, Sanjib Kumar Sharma, François Chappuis, Nicolas Ray.

**Visualization:** Carlos Ochoa.

**Writing – original draft:** Gabriel Alcoba, Carlos Ochoa.

**Writing – review & editing:** Gabriel Alcoba, Carlos Ochoa, Sara Babo Martins, Rafael Ruiz de Castañeda, Isabelle Bolon, Franck Wanda, Eric Comte, Armand Seraphin Nkwescheu, Sanjib Kumar Sharma, François Chappuis, Nicolas Ray.

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
