## [Decision Letter · Decision Letter 0]

31 Jul 2020

Dear Dr Alcoba,

Thank you very much for submitting your manuscript "Development of a transdisciplinary methodology for cross-sectional analysis of snakebite epidemiology at a national scale" for consideration at PLOS Neglected Tropical Diseases. As with all papers reviewed by the journal, your manuscript was reviewed by members of the editorial board and by several independent reviewers. The reviewers appreciated the attention to an important topic. Based on the reviews, we are likely to accept this manuscript for publication, providing that you modify the manuscript according to the review recommendations. 

Sincerely,

M. Abul Faiz

Guest Editor

Jean-Philippe Chippaux

Deputy Editor

Reviewer's Responses to Questions

**Key Review Criteria Required for Acceptance?**

**Methods**

-Are the objectives of the study clearly articulated with a clear testable hypothesis stated?

-Is the study design appropriate to address the stated objectives?

-Is the population clearly described and appropriate for the hypothesis being tested?

-Is the sample size sufficient to ensure adequate power to address the hypothesis being tested?

-Were correct statistical analysis used to support conclusions?

-Are there concerns about ethical or regulatory requirements being met?

Reviewer #1: This manuscript centers in the detailed description of a novel methodological platform proposed to study snakebite from a multi-disciplinary, One Health perspective. Therefore, the description and justification of the methods proposed for this study is a centerpiece of the work. The methodology used is clearly described and adequately justified. The authors developed an original methodological approach, explaining the choices they took in various aspects of the design of the sampling method. The authors provide robust arguments for the selection of a cluster random sampling with the aid of spatial sampling, and present a detailed description of the steps followed in the selection of the various methodological alternatives. 

The detailed description and justification of the methodology chosen represents a valuable contribution for the implementation of similar or related projects in other regions by other groups.

Reviewer #2: This paper provides a reasonably valuable discussion of epidemiological methods but presents no outcome data whatsoever - this doesn't reflect the title.

While the answers are therefore Yes to these questions, I think the paper must include the data generated by the methodologies selected by the authors.

Reviewer #3: The authors address a highly relevant issue through this study. The paper describes , in an elaborate way, the development of new methodology developed for epidemiological study of snakebites using an innovative strategy based on a multi-cluster random survey enhanced by geo-spatial, One Health and health economics components.

The objective of the publication which is to describe this novel methodology to collect large-scale primary data on snakebite in humans and animals is well articulated. The methods considered for selection of clusters and households are discussed in details and the reason for choosing the sampling method is elaborated. Apart from a through search of published literature, the authors have also taken opinion from experts in the field, which is a commendable initiative. 

The validity of any descriptive survey design depends on two variables: The representativeness of the sample to the population and the validity of measurement. These have been discussed in detail as follows.

Representativeness of the Sample to Population: 

Proportionate to Population Cluster Survey is a well validated survey technique to measure events in large population. The strength and weakness of this has been well described in the methodology section. The methodology the investigators adopt, uses 250 clusters which is adequate for the countries involved in this study. 

Households sampled within a cluster: The 50 households sampled per cluster are an adequate number. It is interesting that representativeness has been increased using special geographic techniques in this methodology. This must be appreciated. 

Design Effect: Sample size estimation has one big issue in cluster survey. That is design effect (DE). The investigators have used 1.8 as DE, which may be too small. A DE of at least 3 might have been preferable and 2 would be a constraint value. 

There are no concerns about the ethical or regulatory requirments being met.

**Results**

-Does the analysis presented match the analysis plan?

-Are the results clearly and completely presented?

-Are the figures (Tables, Images) of sufficient quality for clarity?

Reviewer #1: At first, I was expecting to see some results of the community surveys implemented in this study. However, then it became clear that the complexity of the design of the study and the detailed explanation of the building of the methodological platform used had enough strength by itself to justify a publication. Hence, the results basically discuss the various aspects of the design of the methodology. The results section is clear, well presented and c omplete. The tables and figures are clear and of satisfactory quality.

The authors need to be more explicit on why the survey in Cameroon covered most of the country, while in Nepal covered only the Terai region. An explanation is provided in line 370 and thereafter, but it seems insufficient to this reviewer.

Why was the minimal estimated incidence for both countries the same (100 cases per 100,000 pop. per year)?

The inclusion of snakebites in domestic animals is a novelty of this approach and goes along the paradigm of One Health. However, the authors need to consider that this type of survey involves several species of domestic animals, which vary depending on the region. How is this heterogeneity in the type of prevalent domestic animals taken into consideration in the methodology?

Reviewer #2: Not applicable as there were no results.

Reviewer #3: In the results section, the authors describe in great detail, the Sample size calculation, methods chosen for cluster selection, ensuring geographical representativeness , method used for random household selection, and data collection. All the possible methods that were considered, the advantages and disadvantages of each and reason for choosing a particular method based on set criteria is very clearly described. The thought process of the authors is clear to the reader. The use of tables to summarise the relevant data is appreciable . All the tables and figures used are of clear, well labelled and of good quality.

**Conclusions**

-Are the conclusions supported by the data presented?

-Are the limitations of analysis clearly described?

-Do the authors discuss how these data can be helpful to advance our understanding of the topic under study?

-Is public health relevance addressed?

Reviewer #1: The main conclusions tend to support the choices made by the research team concerning the various methodologies required for each aspect of the project. The reationale for such choices is clear and sound.

Reviewer #2: The limited value of this paper is the discussion of the various epidemiological methodologies. 

The authors assertion that their multi-disciplinary methodology enables the most robust national estimates of snakebite incidence remains questionable because no data was provided to compare with other data.

Reviewer #3: The authors draw right conclusions from the data presented in the study . The limitations , although few, are clearly presented. The validity and generalizability are of a satisfactory level and the presented methodological framework would be of use in other high burden countries. The relevance of this novel methodology in the understanding of public health impact of snake bites is discussed well. The authors also discuss how the methodology may be extended to study other neglected tropical diseases.

**Editorial and Data Presentation Modifications?**

Reviewer #1: The manuscript is very well written and clear. No major editorial modifications are recommended.

Reviewer #2: Re the livestock/animal snakebite estimates - it would be interesting to understand how the authors could confidently identify that the animals had been subjected to snakebite as opposed to similar pathologies exerted from other causes (spider/scorpion bites; other pathogens).

Reviewer #3: No suggestions

**Summary and General Comments**

Reviewer #1: This is a novel and valuable contribution, and a timely one, in the light of the current efforts to obtain more accurate data on the incidence and impact of snakebite envenomings on a global and regional basis. The study presents a novel,. sound methodology for assessing snakebite envenoming, with detailed explanations and justifications of the design and choice of the methodology described. This contribution, which takes an inter-disciplinary and One Health perspective, will be valuable for the development of more holistic analyses of the various aspects of snakebites.

Reviewer #2: See above - provide the data gained from using the selected methodologies.

Reviewer #3: The authors address a highly relevant issue through this study. 

In addition to their significant impact on human health in endemic regions, venomous snakebites also inflict a heavy toll on livestock, thereby worsening economic hardships in already impoverished communities. For instance, each year, an estimated 10,000 cattle are envenomed due to Bothrops asper bites in the Central Pacific region of Costa Rica with a 50% fatality rate in the envenomed animals. Since more than 80% of Asian and African rural households depend on livestock and working animals for their livelihood, it is likely that envenoming in these animals may have a significant though under-reported economic impact in snakebite endemic regions. There is an urgent need for large community-based studies focussing on a co-ordinated, collaborative, multidisciplinary and cross-sectoral approach, such as One Health, to address diseases like snakebite envenoming that originate at the animal-human-ecosystems interface.

The current study is an appreciable attempt at devising a methodology to address this knowledge gap satisfactorily, using an innovative strategy based on a multi-cluster random survey enhanced by geospatial, One Health and health economics components. The validity of any descriptive survey design depends on two variables: The representativeness of the sample to the population and the validity of measurement. These have been discussed in detail as follows.

Representativeness of the Sample to Population: 

Proportionate to Population Cluster Survey is a well validated survey technique to measure events in large population. The strength and weakness of this has been well described in the methodology section. The methodology the investigators adopt, uses 250 clusters which is adequate for the countries involved in this study. 

Households sampled within a cluster: The 50 households sampled per cluster are an adequate number. It is interesting that representativeness has been increased using special geographic techniques in this methodology. This must be appreciated. 

Design Effect: Sample size estimation has one big issue in cluster survey. That is design effect (DE). The investigators have used 1.8 as DE, which may be too small. A DE of at least 3 might have been preferable and 2 would be a constraint value. 

Validity of Measurement Tool 

One of the factors that can affect validity of measurement in snakebite here is the verbal recall method that is used to assess incidence of snake bite. Snakebites are incidents that in most cases can be clearly recalled and it is likely that valid information will be obtained using this method. It may be good to get a sub-sample second recall to validate the recall method itself.

The flexibility and easy adaptability of to variable conditions and the use of freely available software and data sources are additional advantages of this study.

In summary, this article is an interesting and novel attempt to address the snakebite epidemiology in two endemic countries using the One Health Approach with an inclusion of geospatial and health economics components. The validity and generalizability are of a satisfactory level and the presented methodological framework would be of use in other high burden countries.

PLOS authors have the option to publish the peer review history of their article (what does this mean?). If published, this will include your full peer review and any attached files.

Reviewer #1: No

Reviewer #2: No

Reviewer #3: No
---

## [Decision Letter · Decision Letter 1]

1 Dec 2020

Dear Dr Alcoba,

We are pleased to inform you that your manuscript 'Novel transdisciplinary methodology for cross-sectional analysis of snakebite epidemiology at a national scale' has been provisionally accepted for publication in PLOS Neglected Tropical Diseases.

Best regards,

M. Abul Faiz

Guest Editor

Jean-Philippe Chippaux

Deputy Editor

Reviewer's Responses to Questions

**Key Review Criteria Required for Acceptance?**

**Methods**

-Are the objectives of the study clearly articulated with a clear testable hypothesis stated?

-Is the study design appropriate to address the stated objectives?

-Is the population clearly described and appropriate for the hypothesis being tested?

-Is the sample size sufficient to ensure adequate power to address the hypothesis being tested?

-Were correct statistical analysis used to support conclusions?

-Are there concerns about ethical or regulatory requirements being met?

Reviewer #1: (No Response)

Reviewer #2: Revisions address all my prior issues

Reviewer #3: No additional comments

**Results**

-Does the analysis presented match the analysis plan?

-Are the results clearly and completely presented?

-Are the figures (Tables, Images) of sufficient quality for clarity?

Reviewer #1: (No Response)

Reviewer #2: Revisions address all my prior issues

Reviewer #3: No additional comments

**Conclusions**

-Are the conclusions supported by the data presented?

-Are the limitations of analysis clearly described?

-Do the authors discuss how these data can be helpful to advance our understanding of the topic under study?

-Is public health relevance addressed?

Reviewer #1: (No Response)

Reviewer #2: Revisions address all my prior issues

Reviewer #3: No additional comments

**Editorial and Data Presentation Modifications?**

Reviewer #1: (No Response)

Reviewer #2: Revisions address all my prior issues

Reviewer #3: No additional comments

**Summary and General Comments**

Reviewer #1: The authors have adequately addressed the issues raised to the first version of this manuscript.

Reviewer #2: Revisions address all my prior issues

Reviewer #3: No additional comments

PLOS authors have the option to publish the peer review history of their article (what does this mean?). If published, this will include your full peer review and any attached files.

Reviewer #1: No

Reviewer #2: No

Reviewer #3: **Yes: **Ravikar Ralph

---

## [Editor Report · Acceptance letter]

8 Feb 2021

Dear Dr Alcoba,

We are delighted to inform you that your manuscript, "Novel transdisciplinary methodology for cross-sectional analysis of snakebite epidemiology at a national scale," has been formally accepted for publication in PLOS Neglected Tropical Diseases.

Best regards,

Shaden Kamhawi

co-Editor-in-Chief

Paul Brindley

co-Editor-in-Chief
